# Evaluation of the Antimicrobial Effect of Graphene Oxide Fiber on Fish Bacteria for Application in Aquaculture Systems

**DOI:** 10.3390/ma15030966

**Published:** 2022-01-26

**Authors:** Ji Hyun Lee, Heejoun Yoo, Yu Jin Ahn, Hyoung Jun Kim, Se Ryun Kwon

**Affiliations:** 1Department of Aquatic Life Medical Sciences, Sunmoon University, Asan 31460, Korea; jihyun804@naver.com; 2Grapheneall, Siheung 15093, Korea; hjyoo1128@gmail.com; 3SamhwanTF, Nowon-gu, Seoul 10848, Korea; yji3210@gmail.com; 4OIE Reference Laboratory for VHS, National Institute of Fisheries Science, Busan 46083, Korea; 5Genome-Based BioIT Convergence Institute, Asan 31460, Korea

**Keywords:** graphene oxide, graphene oxide polyester fiber, fish bacterial disease, water treatment

## Abstract

The growing importance of the domestic aquaculture industry has led not only to its continuous development and expansion but also to an increase in the production of wastewater containing pathogenic microorganisms and antibiotic-resistant bacteria. As the existing water purification facilities have a high initial cost of construction, operation, and maintenance, it is necessary to develop an economical solution. Graphene oxide (GO) is a carbon-based nanomaterial that is easy to manufacture, inexpensive and has excellent antimicrobial properties. In this study, the antimicrobial effect of GO polyester fibers on seven species of fish pathogenic bacteria was analyzed to evaluate their effectiveness in water treatment systems and related products. As a result of incubating GO polyester fibers with seven types of fish pathogenic bacteria for 1, 6, and 12 h, there was no antimicrobial effect in *Vibrio harveyi, V. scopthalmi*, and *Edwardsiella tarda*. In contrast, GO fibers showed antimicrobial effects of more than 99% against *A. hydrophila, S. parauberis, S. iniae*, and *P. piscicola*, suggesting the potential use of GO fibers in water treatment systems.

## 1. Introduction

The aquaculture industry is growing rapidly, along with the domestic aquaculture industry. Most domestic aquaculture farms are gathered in adjacent areas and are designed as a flow-through system, so a large amount of water is discharged. However, the continuous development and expansion of such farms lead to an increase in the production of discharged water containing pathogenic microorganisms, dead fish carcasses, residual antibiotics, and antibiotic-resistant bacteria, thus causing problems because close aquaculture farms are taken and used. Sterilization methods using ultraviolet rays, ions, high-pressure, and high-temperature treatments have been used [1,2,3,4,5]. However, these technologies are inefficient and have high initial construction, operation, and maintenance costs. Therefore, it is necessary to develop a technology that is economically feasible. In addition, mechanical filtration technologies can control aerosols and hydrosols, and fiber filtration systems are highly utilized because of their ability to trap particles and microorganisms. In particular, microfibers and nanofibers provide a chemical-free, cost-effective, and environmentally friendly approach to improve the filtration efficiency and performance of the filtration systems [6,7,8,9,10,11] However, microorganisms trapped in the fibers can survive and multiply [12,13,14,15], thereby causing problems. Therefore, various antibiotics, such as antimicrobial and antiviral agents, were integrated with the filter media to remove the captured pathogenic microorganisms [16,17,18,19]. Moreover, the continuous use of antimicrobial agents can lead to the development of antibiotic-resistant bacteria [20,21], and hence countermeasures are needed to prevent it.

Graphene oxide (GO) is easy to manufacture, inexpensive and has excellent antimicrobial properties. It also has a two-dimensional honeycomb structure that can hold oxygen at the edges and basal planes [22]. The basal plane is composed of epoxy, carbonyl, and hydroxyl groups, and the edge is composed of carboxyl (-COOH) and hydroxyl (-OH) groups, all of which contain hydrophilic structures [22,23]. This facilitates the interaction with biomolecules including lipids, proteins, and DNA, thereby enveloping the bacteria and isolating it from the surrounding environment to limit nutrients and inducing dialysis by promoting the reduction of carbon radicals of GO on the surface of cell membranes, leading to the antimicrobial action [23,24,25,26]. Products that apply the antimicrobial mechanism of GO are already being developed in Korea, but the antimicrobial effect on fish diseases has not been studied sufficiently.

Therefore, in this study, the antimicrobial effect of GO in polyester fibers against *Aeromonas hydrophila*, *Vibrio harveyi*, *V. scophthalmi*, *Streptococcus parauberis*, *S. iniae*, *Edwardsiella tarda*, and *Photobacterium piscicola* (fish bacteria), which continuously cause an economic loss in aquaculture, was analyzed to evaluate its suitability for use in water treatment systems and related products.

## 2. Materials and Methods

### 2.1. Preparation of GO 

The GO used in this study was prepared by adding 10 g of natural graphite powder (under 100 µm) to 900 mL of sulfuric acid (H_2_SO_4_) and 100 mL of phosphoric acid (H_3_PO_4_) and then stirring the mixture at 4 °C using an ice bath. Then, 60 g of potassium permanganate (KMnO_4_) was slowly added and mixed for 1 h. The mixture was then heated to 40 °C for 12 h. Next, 2 kg of ice was poured into the mixture to cool it, and then 20 mL of H_2_O_2_ was added to obtain a GO solution, exfoliated by oxidation. The resulting solution was centrifuged at 10,000 rpm for 10 min to obtain a GO product (crude product) and washed repeatedly with distilled water (DW). The purified GO product was dried in a vacuum for 48 h and prepared as a powder using a fine mill.

### 2.2. Characterization of GO 

A UV spectrophotometer (Orion AquaMate AQ8000, Thermo Scientific, Waltham, MA, USA) was used to measure the optical absorption properties of GO in the 190−800 nm range. An aqueous solution of GO (0.1%) was diluted 10-fold and placed in a quartz cell. A 0.1% aqueous solution of GO was measured using a xenon lamp at 25 °C to analyze the pH and ion conductivity (TOADKK’s WM-32EP model).

A scanning electron microscope (Hitachi Regulus SU-8100) was used to analyze the shape of the GO. Water-diluted GO flakes solution was mixed with methanol (1:5) and applied on the silicon water (previously treated with piranha solution using a dip coater. Samples were dried in a vacuum dryer.

Atomic force microscopy (AFM) analysis was carried out to investigate both the lateral sheet dimensions and sheet thickness, with the intention of discovering how many layers graphene oxide has.

### 2.3. Preparation of GO Polyester Fiber

In this study, polyester fibers containing 0.1% GO were prepared by the extraction method as follows: 100 g of GO powder was suspended in 1% ethanol and added to 1 kg of PET virgin chip. By mixing and drying them, PET virgin chips were coated with GO and extruded at temperatures of 250−280 °C using a twin-screw extruder.

The 1% GO master batch produced was mixed with homo PET and then melt-spun so that the fiber attained a 0.05% GO content. The spun unstretched fiber was stretched in multiple stages and then cut after the crimping process to prepare a short fiber.

Thereon, it could be manufactured according to the pack part (nozzle) in the shape of a solid-type fiber. In this study, we prepared a polyester fiber containing 0.05% GO with a fiber thickness of 3 deniers and length of 51 mm.

### 2.4. Fish Pathogenic Bacteria

In this study, seven pathogenic fish bacteria were used. *S. parauberis*, *S. iniae*, *V. harveyi*, *V. scophthalmi*, and *E. tarda* were kindly provided by Professor Kim, D.H. (Department of Aquatic Life Medicine, Pukyong National University). *A. hydrophila* was obtained from diseased common carp (*Cyprinus carpio*) in our laboratory. *P. piscicola* was isolated from an olive flounder (*Paralichthys olivaceus*).

*A. hydrophila* strains were inoculated on tryptic soy agar (TSA, BD), and TSA containing 1% NaCl was used to cultivate *A. salmonicida*, *V. harveyi*, *V. scophthalmi*, and *E. tarda*. *S. parauberis* and *S. iniae* were cultured on brain heart infusion agar (BHIA, BD) containing 1% NaCl at 27 °C for 24 h. These bacteria were inoculated into each medium (tryptic soy broth or BHIB) and incubated at 27 °C so that the optical density (OD 600) value was 1.0, and the final concentration of all bacteria was adjusted to 1 × 10^8^ CFU/mL for use.

To evaluate the antimicrobial activity of GO polyester fibers, 4 μL of each bacterial culture (*S. parauberis*, *S. iniae*, *V. harveyi*, *V. scophthalmi*, *A. salmonicida*, *E. tarda*, and *Photobacterium*) was inoculated in 40 mL of filtered seawater, and the same amount of *A. hydrophila* bacterial culture solution was inoculated into 40 mL of DW and diluted to a final concentration of 1 × 10^4^ CFU/mL.

After that, 1 g of GO polyester fiber was added and incubated at 27 °C for 1, 6, and 12 h. The supernatant of each time period was diluted 10-fold with phosphate buffer saline (PBS), inoculated on TSA or BHIA, and incubated at 27 °C for 24 h. In the control group, the same method was performed using filtered sterilized seawater or DW, which did not contain GO polyester fibers. Thereafter, the colony was counted to calculate colony-forming units (CFUs).

## 3. Results

### 3.1. Characterization of Graphene Oxide (GO)

Graphene Oxide (GO) was analyzed using an X-ray photoelectron spectroscopy (XPS). Quantitative evaluation in XPS is relatively evaluated using the area ratio of each peak, and it is confirmed that the carbon content is 63.63%, the oxygen content is 35.61%, and the impurity sulfur is less than 1% (Figure 1a). ICP-MS test was also performed to analyze the content of metallic impurities (Appendix A).

For the observation of GO using an electron microscopic, a dispersion of GO was applied on a silicon substrate and dried for scanning electron microscopy (SEM) (Figure 1b). The light-colored part is GO composed of a single layer, and the dark part is GO produced by overlapping with less exfoliation. It was confirmed that most of them consisted of a single layer. AFM images were obtained by contacting the tip with graphene oxide under the condition of 3 nN using the contact mode and scanning an area of 25 µm × 25 µm at a scan rate of 0.5 Hz (Figure 2a). As shown in Figure 2b,c, it was confirmed that graphene oxide has a thickness distribution with ~1.18 nm through AFM analysis and it could be seen that a single layer of graphene oxide was formed.

### 3.2. Antimicrobial Effect of GO Polyester Fiber

The GO polyester fibers used in this study are shown in Figure 3. As a result of treating GO polyester fibers with seven types of fish pathogenic bacteria, the GO fiber was not detected in *A. hydrophila* within 6 h, and *S. parauberis* and *Photobacterium* sp. within 12 h, thereby showing a 100% effect on them. On the other hand, the CFU of the control groups increased after 6 h (Figure 4 and Figure 5).

In addition, *S. iniae* decreased to 5.1 × 10^4^ CFU/mL and 3 × 10^2^ CFU/mL for 1 h and 12 h of incubation, respectively, and these results showed an antimicrobial effect of 99%. However, there was no change in the control group. *V. scophthalmi* showed that the number of bacteria reduced by more than 99% in all the bacterial culture periods of the GO fibers compared to the control group, but there were no further changes. *V. harveyi* and *E. tarda* did not differ from the control group, even after 12 h of incubation (Table 1).

## 4. Discussion

With the growing importance of aquaculture, there is a continuous increase in wastewater discharge. Water treatment or sterilization of wastewater is essential because it includes pathogenic bacteria, viruses, and antibiotic-resistant bacteria and can cause diseases in aquaculture fish. However, the existing water treatment or sterilization methods are inefficient and incur significant costs; therefore, it is necessary to develop an efficient and inexpensive technology.

GO has a high antimicrobial effect at a low cost and has been reported to have a high sterilization effect on *Staphylococcus aureus* (Gram-positive bacteria) and *Pseudomonas aerginosa* (Gram-negative bacteria) [27,28]. In this study, the antimicrobial effect of GO polyester fiber was evaluated using seven types of fish bacterial diseases to analyze its suitability for water treatment and sterilization applications in aquaculture farms.

*A. hydrophila* is a freshwater Gram-negative bacterium that is motile due to its dual-functional flagellar system and causes ulcers, hemorrhagic sepsis, and ascites [29,30]. The genus *Streptococcus* is Gram-positive and non-motile [31]. *S. iniae* was first isolated from freshwater dolphins in 1976 in the United States, and it can infect a large number of freshwater and marine fish, cause hemorrhage, abdominal distension, and ascites [32]. *S. parauberis* was first identified in aquaculture turbot in 1993, and the infected fish showed symptoms such as ascites, congestion of the liver, and enlarged spleen [33]. *Photobacterium piscicola* is a marine Gram-negative bacterium, motile, and the infected fish show a distended stomach, extensive hemorrhage, and petechiae of the gills and liver [34,35]. The genus *Vibrio* is a Gram-negative bacterium, has flagella, and exhibits motility [36,37]. *V. harveyi* is a serious bacterial pathogen that infects a wide range of marine vertebrates and invertebrates, including fish, shrimp, lobsters, and mollusks, and the infected fish showed symptoms such as corneal hemorrhage, vascular and gastrointestinal inflammation [38]. *V. scophthalmi* was first reported in Spain in 1997 and has since developed in the flounder in Asia, and the infected fish showed symptoms such as abdominal distention, liver and intestinal hemorrhage, ascites, spleen and kidney hypertrophy [39,40,41]. *E. tarda* is a Gram-negative pathogen that inhabits a variety of environments and a wide range of hosts and exhibits motility due to its peritrichous flagella [42,43].

In these results, the GO polyester fiber showed an excellent antimicrobial effect of more than 99% against fish pathogenic bacteria, but there was no change in the control group. The antimicrobial effect was not observed in *V. scophthalmi*, *V. harveyi*, and *E. tarda* (Table 1), and it is thought that the bacteria were not captured in the fibers and were not affected by the carbon radical of GO due to the strong flagellar motility of the genera *Vibrio* and *E. tarda*, which have peritrichous flagella. Therefore, for such high-motility bacteria, it is necessary to analyze the antimicrobial effect of other composite graphene fibers and develop a new complementary method. In addition, as the maximum contact time of bacterial culture in this study was 12 h, if the contact time with GO is increased to continue its effect, or a high concentration of GO is used, there is a possibility of an antimicrobial effect on high-motility bacteria as well, but further complementary experiments are necessary to evaluate this. In contrast, *A. hydrophila* with flagella showed an antimicrobial effect of 100% (Table 1, Figure 4). Bacterial flagella play an important role in motility and chemotaxis and some studies have reported that they are also involved in host adhesion or evasion [44,45]. *A. hydrophila* is a monotrichous bacterium that is involved in host adhesion [46] because it has weaker motility than the genus *Vibrio*, lophotrichous bacteria, and *E. tarda*, peritrichous flagella, and it cannot be evaded, and the bacteria are captured in the fibers and GO, which is thought to have an antimicrobial effect. In addition, GO fibers showed an antimicrobial effect of more than 99% in non-motile pathogenic bacteria. These results are the first to quantify the antimicrobial action on fish bacterial diseases using GO, and this study suggests the possibility that GO can be applied to water treatment systems or breeding tanks.

## Figures and Tables

**Figure 1 materials-15-00966-f001:**
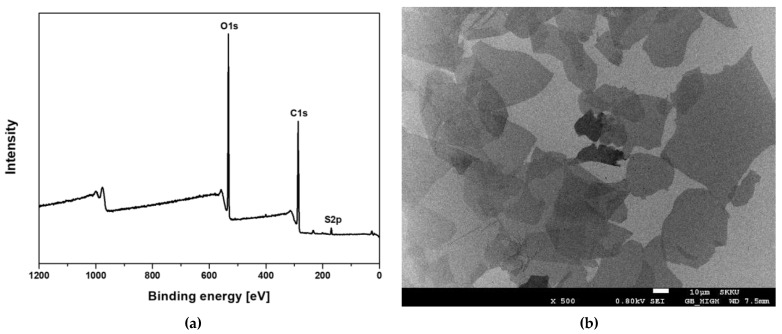
Analysis of the graphene oxide by (**a**) X-ray photoelectron spectroscopy (XPS) and (**b**) scanning electron microscope (SEM).

**Figure 2 materials-15-00966-f002:**
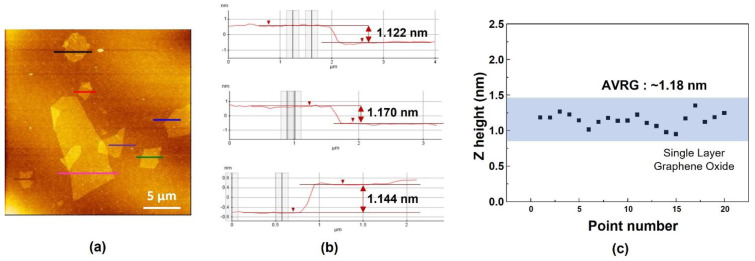
The line profile measured from the AFM image and its thickness distribution. (**a**) the AFM image, (**b**) thickness distribution of graphene oxide, (**c**) the height distribution of graphene oxide.

**Figure 3 materials-15-00966-f003:**
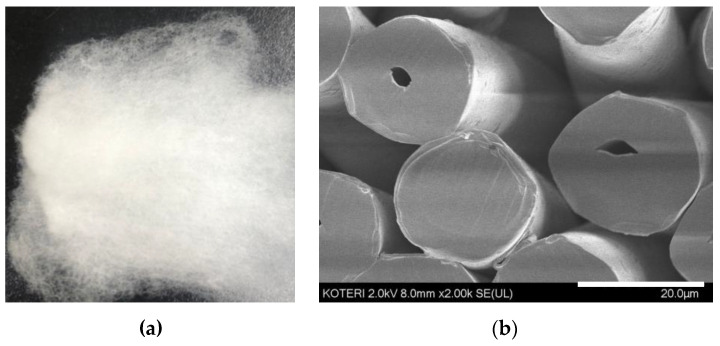
Graphene oxide polyester fiber (**a**) and its cross-sectional shape (**b**).

**Figure 4 materials-15-00966-f004:**
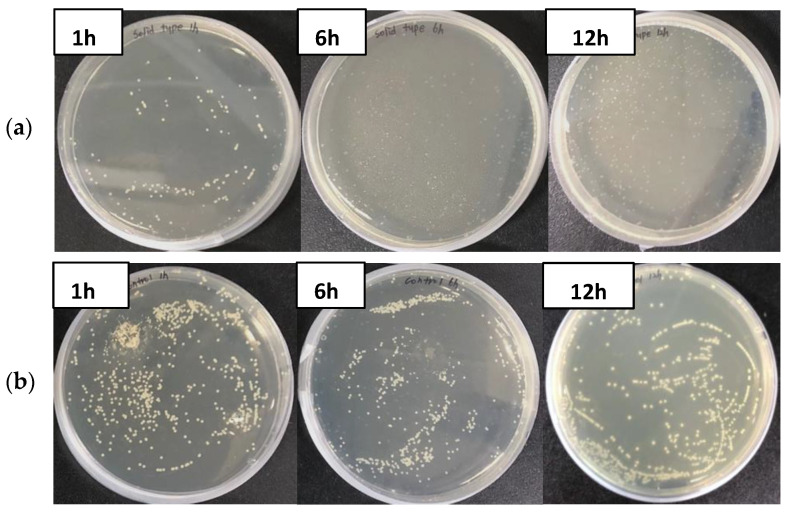
Growth of *Aeromonas hydrophila* on TSA plates incubated with graphene oxide polyester fiber for 1, 6, and 12 h. (**a**) GO fiber, (**b**) control.

**Figure 5 materials-15-00966-f005:**
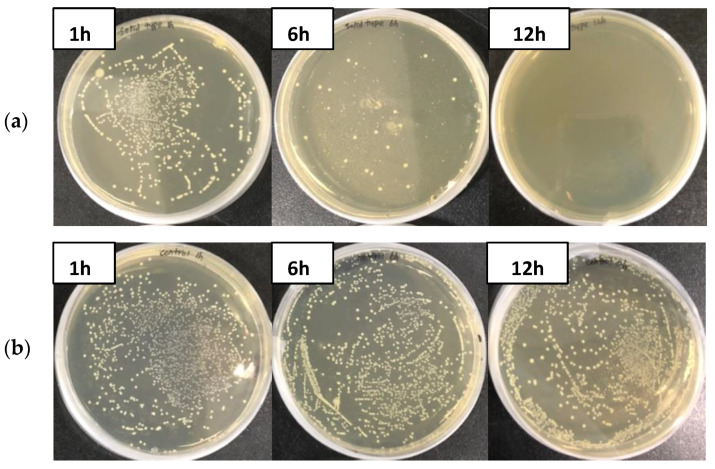
Growth of *Streptococcus parauberis* on TSA plates incubated with graphene oxide polyester fiber for 1, 6 and 12 h. (**a**) GO fiber, (**b**) control.

**Table 1 materials-15-00966-t001:** Antimicrobial effect of graphene oxide polyester fiber to fish bacteria.

Bacteria		1 h (CFU/mL)	6 h (CFU/mL)	12 h (CFU/mL)	Antimicrobial Effect (%)
*A. hydrophila*	GO fiber	1.17 × 10^4^	0	0	100
	Control	5.6 × 10^4^	4.5 × 10^4^	6.3 × 10^4^	0
*S. parauberis*	GO fiber	6.3 × 10^4^	1.6 × 10^3^	0	100
	Control	1.23 × 10^5^	1.24 × 10^5^	1.2 × 10^5^	0
*S. iniae*	GO fiber	5.1 × 10^4^	1 × 10^3^	3 × 10^2^	99
	Control	4.3 × 10^4^	8.5 × 10^4^	5.7 × 10^4^	0
*P. piscicola*	GO fiber	5.5 × 10^4^	1.9 × 10^2^	0	100
	Control	9.2 × 10^4^	5 × 10^4^	1 × 10^5^	0
*E. tarda*	GO fiber	9.2 × 10^4^	8 × 10^4^	8.8 × 10^4^	0
	Control	1 × 10^5^	1 × 10^5^	5.6 × 10^4^	0
*V. scophthalmi*	GO fiber	6 × 10^3^	4.4 × 10^3^	9.2 × 10^3^	0
	Control	2.6 × 10^5^	1 × 10^5^	1 × 10^5^	0
*V. harveyi*	GO fiber	7 × 10^4^	2 × 10^5^	2.5 × 10^5^	0
	Control	5.2 × 10^4^	1.5 × 10^5^	2.2 × 10^5^	0

## Data Availability

Data are contained within the article and Appendix A.

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
