# Peer review of "Evaluation of the Antimicrobial Effect of Graphene Oxide Fiber on Fish Bacteria for Application in Aquaculture Systems"

_materials, 2022, doi:10.3390/ma15030966_

Round 1
Reviewer 1 Report
The study interestingly shows the antimicrobial effects of a GO-composite material on many types of bacteria.
The study is well performed, although, a few things need to be improved.
- Discuss the differences between the possible antimicrobial effect and capturing effect. These two effects are different and should be discussed so that the reader knows what the proposed explanation is. Since it is not clear what factors make one form antimicrobial and the other form not.
- GO-composite fibers should be better characterized. This might lead to different conclusions when compared with the biological effects. Currently, we only have the photograph of the fibers, and the knowledge that one of the fibers were coated with silicon. It is difficult to draw conclusions of why the phenomenon is occurring with this limited amount of evidence.
- Methods: THe silicon coating step is missing. This should also be described in greater detail.
- Figure 1b resolution/scale bar
- Figure 2 scale bars/resolution in 2a
- explanation in lines139-142 need to be clarified.
Author Response
Thank you very much for your kind review and useful comments.
I revised the MS followed by your suggestions as follows.
- Discuss the differences between the possible antimicrobial effect and capturing effect. These two effects are different and should be discussed so that the reader knows what the proposed explanation is. Since it is not clear what factors make one form antimicrobial and the other form not.
→ Thanks for the good question. If the decrease in the number of bacteria confirmed in this study was due to the capturing effect of the fibers, the decrease in the number of bacteria would have occurred in all bacteria. However, in E.tarda, V. scophthalmi, and V. harveyi, there was no change or a slight increase in the number of bacteria, so it is determined to be an antibacterial action by graphene.
Since the hollow type fiber used in this study had silicone treatment on the fiber as well as the holes in the fiber, it is not easy to prove whether it is because of the silicone or the fiber type that the antibacterial effect was not shown in the hollow type fiber. Therefore, in this manuscript, we decided to focus on solid type GO fiber showing different antibacterial effects depending on the type of bacteria.
Therefore, all contents of silicone coating applied to hollow type fiber were deleted.
- Methods: The silicon coating step is missing. This should also be described in greater detail.
→ Thanks for your comment. In this manuscript, we decided to focus on solid type GO fiber showing different antibacterial effects depending on the type of bacteria. Therefore, all contents of silicone coating applied to hollow type fiber were deleted
- Explanation in lines 139-142 need to be clarified.
→ Thanks for your suggestion. According to your suggestion, the SEM sample preparation process was described in M & M (line 82) as follows: “Water-diluted GO flakes solution was mixed with methanol (1:5) and applied on the silicon water (previously treated with piranha solution using a dip coater. Samples were dried in a vacuum dryer”.
1) Figure 1b resolution/scale bar
2) Figure 2 scale bars/resolution in 2a
→ About 2. According to your suggestion, we replaced the figure and added a scale bar in Fig 1~Fig 3.
Thank you again for your comments.

Reviewer 2 Report
The manuscript entitled „Evaluation of the antimicrobial effect of graphene oxide fiber on fish bacteria for application in aquaculture systems” denotes scientific attempts towards establishing the antimicrobial influence of composite polyester–based fibers containing GO flakes on fish bacterial diseases. In my opinion, the subject of the study is interesting, however Authors of the manuscript will have to address several issues prior to publication of their manuscript:
- Regarding the final application, it would be interesting to note how pure was the GO material, to be sure that the antimicrobial properties were driven by the material itself, not by the toxic, post-production impurities. Did the Authors check the chemical composition of GO flakes after the synthesis?
- Did the samples were coated with conducting agent before SEM imaging? Please add comments regarding this issue in section 2.2 of the manuscript.
- Did the Authors check the agglomeration of GO flakes within the fibers? It would be beneficial to ensure homogenous distribution of carbon-based phase within the whole volume of the fibers.
- Regarding sizes of the obtained GO flakes, after implementation of fine milling as the final stage of the material preparation protocol I would expect to see smaller sizes than graphite precursor. Please comment on that and in addition please specify what was the distribution of the size of flakes?
- The Authors claimed that: ”The light-colored part is GO composed of a single layer, and the dark part is GO produced by overlapping with less exfoliation. It was confirmed that most of them consisted of a single layer (Fig. 1b).” (Page 4, Lines 134-137). I would be more than careful when discussing GO flakes thickness based on SEM images. The Authors should implement AFM microscopy in order to confirm their bold thesis, like for instance, in this manuscript:
Chlanda et al.: “Morphology and Chemical Purity of Water Suspension of Graphene Oxide FLAKES Aged for 14 Months in Ambient Conditions. A Preliminary Study”; Materials, 2021
- Regarding Figure 2 it would be nice to see a scalebar.
- In addition please specify what was the reason to confront solid fibers of circular shape against hollow-type star shape fibers. By doing this, the Authors implemented many different factors affecting the final results.
- In addition to aforementioned issue, please clarify why were the hollow-type fibers covered with silicon and the solid type not? I guess that you wanted to take advantage of antimicrobial properties of silicon, but surprisingly you did not. Comment on why did it happened? Moreover, covering fibers with silicon prevents the use of GO flakes benefits as they were covered with silicon too.
- Finally the Authors claimed that: “In the case of the hollow-type fiber, it is thought that the bacteria were not effectively captured because of the smooth surface due to silicon treatment, whereas the solid-type fiber was thought to have effectively captured bacteria because its surface was relatively less smooth.” Do the Authors have any results to support this thesis. I did not find any single image or data covering this issue. Speculating about surface roughness without data obtained with AFM or profilometer is not good idea. I suggest to run additional examination of the obtained material or to rewrite this sentence.
Author Response
Thank you very much for your kind review and useful comments.
I revised the MS followed by your suggestions as follows.
- Regarding the final application, it would be interesting to note how pure was the GO material, to be sure that the antimicrobial properties were driven by the material itself, not by the toxic, post-production impurities. Did the Authors check the chemical composition of GO flakes after the synthesis?
→ According to your suggestion, ICP-MS analysis was performed to analyze the chemical composition of GO flake, and the analysis results are described in ‘Supporting information’.
- Did the samples were coated with conducting agent before SEM imaging? Please add comments regarding this issue in section 2.2 of the manuscript.
→ Thanks for your comment. SEM sample preparation process was described in Materials and Methods (line 82) as follows: “Water-diluted GO flakes solution was mixed with methanol (1:5) and applied on the silicon water (previously treated with piranha solution using a dip coater. Samples were dried in a vacuum dryer”
- Did the Authors check the agglomeration of GO flakes within the fibers? It would be beneficial to ensure homogenous distribution of carbon-based phase within the whole volume of the fibers.
→ Thanks for your comment. It is difficult to confirm the distribution of GO flakes by the optical or electron microscopy when processed with fibers. In addition, since both graphene and the base polymer contain C, EDS analysis is also difficult. Therefore, it is necessary to judge the presence or absence of agglomeration according to the uniformity and color during fiber spinning, but this was not confirmed in this study.
- Regarding sizes of the obtained GO flakes, after implementation of fine milling as the final stage of the material preparation protocol I would expect to see smaller sizes than graphite precursor. Please comment on that and in addition please specify what was the distribution of the size of flakes?
The Authors claimed that: “The light-colored part is GO composed of a single layer, and the dark part is GO produced by overlapping with less exfoliation. It was confirmed that most of them consisted of a single layer (Fig. 1b).” (Page 4, Lines 134-137). I would be more than careful when discussing GO flakes thickness based on SEM images. The Authors should implement AFM microscopy in order to confirm their bold thesis, like for instance, in this manuscript:
Chlanda et al.: “Morphology and Chemical Purity of Water Suspension of Graphene Oxide FLAKES Aged for 14 Months in Ambient Conditions. A Preliminary Study”; Materials, 2021
→ Thanks for your comment. According to your suggestion, the results of AFM analysis are presented in Fig. 2 and are described in line 134 as follows: “The light-colored part is GO composed of a single layer, and the dark part is GO produced by overlapping with less exfoliation. It was confirmed that most of them consisted of a single layer. AFM images were obtained by contacting the tip with graphene oxide under the condition of 3nN using the contact mode and scanning an area of 25 um x 25 um at a scan rate of 0.5 Hz (Fig. 2a). As shown in Fig. 2b and Fig. 2c, it was confirmed that graphene oxide has a thickness distribution with ~ 1.18nm through AFM analysis and it could be seen that a single layer of graphene oxide was formed”.
- Regarding Figure 2 it would be nice to see a scale bar.
→ According to your suggestion, we replaced the figure and added a scale bar.
- In addition to aforementioned issue, please clarify why were the hollow-type fibers covered with silicon and the solid type not? I guess that you wanted to take advantage of antimicrobial properties of silicon, but surprisingly you did not. Comment on why did it happened? Moreover, covering fibers with silicon prevents the use of GO flakes benefits as they were covered with silicon too.
Finally the Authors claimed that: “In the case of the hollow-type fiber, it is thought that the bacteria were not effectively captured because of the smooth surface due to silicon treatment, whereas the solid-type fiber was thought to have effectively captured bacteria because its surface was relatively less smooth.” Do the Authors have any results to support this thesis. I did not find any single image or data covering this issue. Speculating about surface roughness without data obtained with AFM or profilometer is not good idea. I suggest to run additional examination of the obtained material or to rewrite this sentence.
→ Thanks for the good question. Since the hollow type fiber used in this study had silicone treatment on the fiber as well as the holes in the fiber, it is not easy to prove whether it is because of the silicone or the fiber type that the antibacterial effect was not shown in the hollow type fiber. Therefore, in this manuscript, we decided to focus on solid type GO fiber showing different antibacterial effects depending on the type of bacteria. Therefore, all contents of silicone-coated hollow type fiber were deleted in this manuscript.
Thank you again for your comments.

Round 2
Reviewer 1 Report
The authors have satisfactorily addressed the comments
Reviewer 2 Report
The Authors have addressed all the issues raised by this reviewer. I'm satisfied with the work their done and I have no more issues to disclose regarding publication of this manuscript.